# The Karp Dataset

**Mason DiCicco**
Department of Computer Science
Worcester Polytechnic Institute
Worcester, MA 01609
mtdicicco@wpi.edu

**Eamon Worden**
Department of Computer Science
Worcester Polytechnic Institute
Worcester, MA 01609
eaworden@wpi.edu

**Conner Olsen**
Department of Computer Science
Worcester Polytechnic Institute
Worcester, MA 01609
caolsen@wpi.edu

**Nikhil Gangaram**
Department of Computer Science
Worcester Polytechnic Institute
Worcester, MA 01609
nrgangaram@wpi.edu

**Daniel Reichman**
Department of Computer Science
Worcester Polytechnic Institute
Worcester, MA 01609
dreichman@wpi.edu

**Neil Heffernan**
Department of Computer Science
Worcester Polytechnic Institute
Worcester, MA 01609
nth@wpi.edu

## Abstract

Understanding the mathematical reasoning capabilities of Large Language Models (LLMs) is a central topic in the study of artificial intelligence. This new domain necessitates the creation of *datasets of reasoning tasks* for both training and benchmarking the performance of LLMs. To this end, we introduce the *Karp dataset*: The first dataset composed of detailed proofs of NP-completeness reductions. The reductions vary in difficulty, ranging from simple exercises of undergraduate courses to more challenging reductions from academic papers. We compare the performance of state-of-the-art models on this task and demonstrate the effect of fine-tuning with the Karp dataset on reasoning capacity.

## 1 Introduction

Perhaps the concept receiving the most attention in theoretical computer science is that of a *reduction*. Loosely speaking, a reduction between decision problems $A$ and $B$ is a mapping $f$ such that: If $x$ is an input to $A$, then $f(x)$ is an input to $B$, and the answer to $x$ is "yes" if and only if the answer to $f(x)$ is "yes." Efficiently computable reductions can be used to leverage algorithms that solve $B$ in order to solve $A$. Furthermore, efficient reductions can establish hardness results: if $A$ is believed to be intractable, and $A$ reduces to $B$ efficiently, then $B$ is intractable as well, since an efficient algorithm for $B$ can be used to solve $A$. This simple observation is at the core of the theory of NP-completeness, which is the topic of thousands of papers and an influential monograph Garey and Johnson [1979].

Our goal is to study the capabilities of Large Language Models (LLMs) and their potential to influence formal mathematics. To that end, we built a new dataset of 90 NP-hardness proofs (reductions) to be used for evaluation and training of language models. We are not aware of the study of LLMs for proving new NP-hardness results (by constructing reductions) or reproving and verifying known results. We believe that aiming language models at reductions *in particular* has great potential to

benefit our understanding of their reasoning capabilities and applicability to formal mathematics. This is because:

- Finding a reduction between two problems is a high-level reasoning task. Imbuing LLMs with the ability to construct reductions could lead to improved reasoning capabilities.

- It is feasible to construct dozens of examples of reductions that are theoretically interesting, go beyond symbolic manipulations to prove mathematical identities, and have a short (several paragraphs) proof using natural language. The existence of short yet difficult-to-find proofs hints that such proofs can be found automatically with reasonable computing resources (e.g., memory, training time).

- Such datasets are challenging to construct in other mathematical domains. Current datasets of mathematical problems (e.g., Hendrycks et al. [2021]) that are used to evaluate math capabilities of large language models generally focus on a single numerical or symbolic outcome.

## 1.1 Related work

There has been extensive recent research directed toward using generative AI, neural networks, and Interactive Theorem Provers (ITP) in pushing the boundaries of mathematics [Azerbayev et al., 2021, Buzzard, 2020, Hendrycks et al., 2021, Lample et al., 2022, Polu et al., 2022, Szegedy, 2020] including proving new theorems as well as reproving known theorems. To our knowledge, they do not include proofs of NP completeness using reductions. Very few works seem to have studied automatically constructing reductions toward establishing NP-completeness results. One of the more advanced datasets similar to ours is The CLRS Algorithmic Reasoning Benchmark of Veličković et al. [2022], which predicts the trajectories of various algorithms using an algorithmic model but explicitly avoids NP-Hard problems. Motivated by the education domain, Creus et al. [2014] study the problem of testing the correctness of reductions using SAT-solvers and designated programming language REDNP to establish NP-completeness. One bottleneck noted in proof verification using SAT solvers is the large size of SAT formulas obtained in the process of verification. Recently, Zhang et al. [2022] introduced Karp, a language for programming and testing reductions, motivated by the educational domain as well. Karp is a Racket-esque framework that can be used to define computational problems as well as reductions between them. In addition to providing a systematic way to construct reductions, Karp automatically tests the correctness of reductions. The Karp dataset contains significantly fewer solved questions compared to most math datasets. It does not use generative AI to find reductions and their proofs.

Related datasets such as MATH [Hendrycks et al., 2020], MathQA [Amini et al., 2019], GSM8K [Cobbe et al., 2021], MGSM [Shi et al., 2022], ProofWriter [Tafjord et al., 2020] and others have offered new ways to evaluate the mathematical reasoning and proof generation capabilities of language models. The MATH dataset consists of challenging problems taken from high school math competitions, testing a model's elementary problem-solving skills across various domains of mathematics. The GSM8K and MGSM (multilingual GSM8K) datasets focus on grade-school math problems, assessing the model's ability to perform arithmetic reasoning and handle multi-step calculations. ProofWriter evaluates a model's proficiency in generating natural language proofs for elementary logical inference tasks, emphasizing multi-hop reasoning. While these datasets are instrumental in testing general mathematical and logical reasoning, they are completely disjoint from the task of constructing reductions for NP-completeness proofs. Reductions in computational complexity involve a unique blend of algorithmic thinking, formal proof techniques, and an understanding of computational problems' intrinsic properties. This gap highlights the need for specialized resources.

## 1.2 Evaluating large language models on the Karp dataset

Datasets such as MATH and MGSM are valuable because they allow for standardized comparison of the capabilities of language models, but LLMs now excel at scoring highly on them. For instance, GPT-4o [Achiam et al., 2023] scores over 90% on GSM8K, 75% on the MATH dataset, and around 86% on MMLU (Massive Multitask Language Understanding) [Hendrycks et al., 2020]. While impressive, there are concerns that LLMs have been overfit on the testing datasets due to their availability on the internet. Moreover, achieving a high level of performance on GSM8K, which consists of grade-school math problems, only indicates that LLMs are comparable to highly skilled

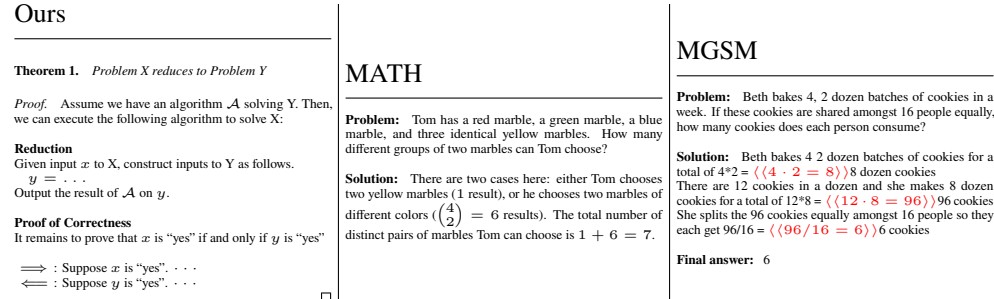

Figure 1: Our reduction template (left) compared to MATH (middle) and GSM8k (right)

eighth graders. As more advanced LLMs such as Strawberry (also known as o1) are released, researchers will be aiming towards matching the problem-solving capacity of undergraduate or even PhD level students. This necessitates datasets of complex higher-education-level questions such as reductions.

## 2 The Karp dataset

Our dataset consists of detailed natural language descriptions of dozens of reductions establishing NP-hardness proofs. These proofs are significantly more involved and labor-intensive to generate relative to math problems with a numerical answer [Hendrycks et al., 2021] or a sequence of computational steps as a solution [Cobbe et al., 2021]. Every reduction in the dataset is sourced from well-known literature such as Garey and Johnson [1979], Papadimitriou [1994], Dasgupta et al. [2006]. The dataset also contains natural language versions of Karp's 21 original NP-complete problems [Karp, 2010]. Other sources include academic papers Garey et al. [1974, 1976], Fomin et al. [2013], Aloise et al. [2009] and dedicated surveys of NP-completeness Ausiello et al. [2012] and the references therein.

Many proofs of NP-completeness in the literature compress proofs of claims that are somewhat tedious to prove formally, and it has been observed that some proofs contain inaccuracies [Zhang et al., 2022]. In our proofs, we attempted to avoid including unproven claims, emphasizing clarity at the cost of verbosity. Such proofs also often rely on diagrams, which we convert to natural language for LLM comprehension. As a result, the proofs in our dataset are somewhat longer than proofs in other datasets, altogether spanning over 170 pages. We avoided including problems with highly complex proofs that require more than two pages. The reductions in the dataset have lengths between 1000 and 6000 characters and have an average length of approximately 2000 characters. The distribution of lengths is depicted in Figure 2. Some examples of reductions can be found in Appendix D, and the full lists of problems and reductions can be found in Tables 5 and 6, respectively. *We will share the full dataset with interested researchers upon request.*

**Formatting**    The dataset consists of reductions (in the form of LaTeX-typeset theorems) between computational problems whose definitions are also provided. Reductions in the dataset adhere to a highly structured template: A precise definition of the mapping followed by a proof of correctness (See Figure 1). The language is fairly expository and instructive: While all the content of a formal proof is present, we frequently include conceptual justification of non-trivial logical steps.

**Omitted details**    In all of our proofs, we omit a key concept needed to establish NP-completeness: Polynomial-time computability and verification. For example, in a proper NP-completeness proof, the mapping from one decision problem to another must be possible to implement efficiently[1], otherwise the reduction is vacuous (e.g., if exponential time is allowed, then one could just brute-force the answer to the original problem.) Efficiency of a reduction is often easy (but tedious) to prove, and we maintain that this holds true for all problems in our dataset. Hence, we choose to mask these details.

---

[1]Here, "efficiently" means "in polynomial time", with respect to the size of the input

| Benchmark | Strawberry | Llama | LlamaReduce |
| --- | --- | --- | --- |
| Test set | 1.5 | 0.875 | 1.25 |
| Challenge set | 0.875 | 0.375 | 0.5 |

Table 1: Average scores achieved by Strawberry, Llama, and LlamaReduce on the two problem sets. In the second row, LlamaReduce has been fine-tuned on the entire Karp dataset, while in the first row, the test set is held out during training.

# 3 Experiments

In contrast to computations and formal logical deductions, natural-language mathematical proofs resist straightforward automatic verification. Due to this limitation, all models are manually evaluated on a small, fixed test set by a human expert (a graduate student in theoretical computer science).

**Test set** We initially evaluated our models on a randomly chosen set of 8 reductions from the dataset, at the level of undergraduate homework assignments (test set). After our initial evaluation, Strawberry was released and achieved significantly better results on the test set. To gain a better understanding of the capabilities of Strawberry, we constructed an additional list of eight more challenging reductions (challenge set) that did not belong to the original dataset.

**Prompts** Models are evaluated on their responses to a highly structured prompt, which asks for a reduction between two decision problems. The prompt provides a LaTeX template for the reduction, which matches the format of the dataset, states the two problems and any necessary definitions, and asks for a detailed reduction. Full examples of prompts can be found in Appendix E.

**Scoring** Completed reductions receive a score of 0, 1, or 2, where 0 represents a completely incorrect answer, 1 reflects a construction that contains significant yet fixable flaws, and 2 indicates a fully or nearly correct reduction with only minor errors. If the response contains superficial bugs (such as LaTeX-compilation errors), we repair these and proceed with normal scoring.

**Models** We compare the performance of OpenAI's recent Strawberry model, the Llama70B-Instruct base model [Touvron et al., 2023] as well as our fine-tuned Llama70B-Instruct model, which we call LlamaReduce. The fine-tuning method we used is described in Appendix B.

**Results** Strawberry achieves impressive averages of 1.5 on the test set, and 0.875 on the challenge set. Interestingly, Strawberry even gave a more compact version of a current well-known reduction in the challenge set (See Appendix E). This outperforms the base Llama model, which scores 0.875 on the test set and 0.375 on the challenge set. The only problem that Llama answered correctly from the challenge set was *NAE4SAT to Set Splitting*, whose difficulty is relatively low. LlamaReduce clearly benefited from fine-tuning on the Karp dataset, as it was able to score 1.25 and 0.5 on the test and challenge sets respectively. The complete breakdown of scores is compiled in Tables 2 and 3 in Appendix C.

These preliminary findings, especially the low scores achieved on the challenge set, suggest that reductions are a challenging task for LLMs, leaving room for potential improvement. For easier reductions (such as those in the test set), fine-tuning was beneficial in improving performance. The impressive performance of Strawberry provides additional evidence that prompt engineering has a significant effect on problem-solving capacity, particularly on problems from the test set (at the level of homework questions from an undergraduate course covering NP-completeness). Both prompt engineering and fine-tuning appear to be less effective for improving performance for the harder reductions such as those in the challenge dataset.

We also evaluate LlamaReduce on the MATH and MGSM datasets. Results are in Appendix C.

| Problem | Strawberry | Llama | LlamaReduce |
|---|---|---|---|
| 3Coloring to Planar 3Coloring | 1 | 0 | 0 |
| 3SAT to Independent Set | 2 | 1 | 1 |
| 3SAT to NAE4SAT | 1 | 0 | 2 |
| Hamiltonian Path to K-SpanningTree | 0 | 0 | 0 |
| Independent Set to Set Packing | 2 | 1 | 2 |
| Independent Set to Vertex Cover | 2 | 2 | 1 |
| Partition to Bin Packing | 2 | 2 | 2 |
| Partition to Knapsack | 2 | 1 | 2 |
| **Average** | 1.5 | 0.875 | 1.25 |

Table 2: Scores achieved by each model on each problem in the test set.

## 4   Conclusion

We have constructed the Karp dataset consisting of reductions establishing NP-completeness. Future work could examine extending the dataset with additional reductions (e.g., reductions establishing hardness of approximation of NP-hard optimization problems Arora et al. [1998], Feige et al. [1996], Dinur [2007]). Using the Karp dataset as well as generative AI more broadly to discover new reductions and simplify known NP-completeness proofs is an exciting future direction.

The lack of automatic verification for natural language proofs of NP-completeness is a bottleneck in creating a larger dataset. In our experiments, language models failed to judge the correctness of reductions. We suspect that a transformation from natural language to more structured representations (e.g., code, formal math, the Karp language) is a required step to allow automatic verification.

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

## A   Test sets

**Test set**   The test set consists of the reductions: *Partition to Knapsack; Independent Set to Set Packing; Independent Set to Vertex Cover; Independent Set to Undirected Feedback Set; Partition to Bin Packing; Clique to Dense Subgraph; Unweighted Max Bisection to Weighted Bisection Width; Hamiltonian Cycle to Hamiltonian Path.*

**Challenge set**   The challenge set consists of the reductions: *NAE4SAT to Set Splitting; Clique to Balanced Biclique; Independent Set to Induced Matching; 3SAT to Contagious Set; 3SAT to Edge Disjoint Paths; 3Coloring to Low Diameter Clustering; Densest Cut to Sum of Squares Clustering; Vertex Cover to Planar Vertex Cover.*

## B   Fine-tuning

We fine-tuned Llama 70B-Instruct using Unsloth. For training, we utilized the AdamW optimizer Loshchilov [2017] and QLora Dettmers et al. [2024] with 4-bit precision to reduce memory consumption. The learning rate was set to $2 \times 10^{-5}$, following a linear scheduler with 10 warmup steps. We applied weight decay of 0.01 to prevent overfitting. The model was trained with a batch size of 8 per device. We used 16-bit floating point precision and random seed 0. LlamaReduce was trained on 1 A100 GPU until the loss converged on a validation set at 10 epochs. All models, fine-tuned or not, were inferenced with a temperature of 0.

## C   Results

This section contains tables of results that were omitted due to space constraints.

| Problem | Strawberry | Llama | LlamaReduce |
|---|---|---|---|
| 3Coloring to Low Diameter Clustering | 2 | 1 | 2 |
| 3SAT to Contagious Set | 0 | 0 | 0 |
| 3SAT to Edge-Disjoint Paths | 1 | 0 | 0 |
| Clique to Balanced Biclique | 0 | 0 | 0 |
| Densest Cut to Sum of Squares Clustering | 0 | 0 | 0 |
| Independent Set to Induced Matching | 1 | 0 | 0 |
| NAE4SAT to Set Splitting | 2 | 2 | 2 |
| Vertex Cover to Planar Vertex Cover | 1 | 0 | 0 |
| Average | 0.875 | 0.375 | 0.5 |

Table 3: Scores achieved by each model on each problem in the challenge set.

| Benchmark | Strawberry | Llama | LlamaReduce |
|---|---|---|---|
| MATH | 85.5 | 68.0 | 68.5 |
| MGSM | 90.8 | 86.9 | 64.5 |

Table 4: Accuracy of Strawberry, Llama, and LlamaReduce on the MATH and MGSM benchmarks.

## D   Examples of reductions

This section contains the reductions *3SAT to Independent Set* as well as *Hamiltonian Path to Bounded-Degree Spanning Tree* as they appear in the dataset.

**3SAT to Independent Set**

**Definition 1.** A 3-*CNF* is a Boolean formula equal to an AND of clauses, where each clause is an OR of exactly 3 literals (i.e., variables or their negations). A 3-CNF is *satisfiable* if there exists an assignment of variables to true (1) or false (0) such that the entire formula evaluates to true.

**Problem 1** (3SAT).

- **Input:** $(X, C)$, where $X = \{x_1, \cdots, x_n\}$ is a set of variables and $C = \{C_1, \cdots, C_m\}$ is a set of clauses containing exactly 3 literals derived from $X$ (i.e., $x_i$ or $\neg x_i$).
- **Output:** $\begin{cases} 1 & \text{There exists an assignment (of variables in } X\text{) satisfying } \phi = C_1 \wedge \cdots \wedge C_m. \\ 0 & \text{Otherwise} \end{cases}$

**Definition 2.** Given an undirected graph $G = (V, E)$, a subset of vertices $S \subseteq V$ is an *independent set* if no two nodes are joined by an edge:

$$\forall u, v \in S : (u, v) \notin E.$$

**Problem 2** (Independent Set).

- **Input:** $(G, k)$ where
  - $G = (V, E)$ is an undirected graph
  - $k$ is a positive integer
- **Output:** $\begin{cases} 1 & G \text{ has an independent set of size } k \\ 0 & \text{otherwise} \end{cases}$

**Theorem 2.** *3SAT reduces to Independent Set*

*Proof.* Assume we have an algorithm $\mathcal{A}$ solving Independent Set. Then, we can execute the following algorithm to solve 3SAT:

**Reduction** Given inputs $(X, C)$ to 3SAT, construct inputs to Independent Set $(G, k)$ as follows:

1. For each clause $C_i = (a_i \vee b_i \vee c_i)$, create a "cluster" of vertices $a_i, b_i, c_i$ in $V$, and connect them in a triangle by adding edges $(a_i, b_i), (b_i, c_i), (c_i, a_i)$ to $E$.
2. Additionally, connect every two vertices corresponding to complementary literals (i.e. there is an edge between every $x_i$ and $\neg x_i$).

Output the result of $\mathcal{A}$ on $(G, k)$, where $k = |C|$.

**Proof of Correctness** To establish correctness, it remains to prove that $\phi$ is satisfiable $\iff$ $G$ has an independent set of size $k$.

$\implies$ : Let $T$ be an assignment of variables satisfying $\phi$. In particular, each clause $C_i$ contains at least one true literal. Construct a set $I$ which contains one such true literal from each clause. We now claim that $I$ corresponds to an independent set in $G$ of size $k$: It contains one vertex (literal) from each of the $k$ clauses, and no pair of vertices in $I$ are adjacent since there is only one vertex per cluster and vertices corresponding to complementary literals (i.e. $x$ and $\neg x$) cannot both be in $I$ since that would be an impossible assignment; $x$ and $\neg x$ cannot simultaneously be true.

$\impliedby$ : Let $I$ be an independent set of size $k$ in $G$. Note that $I$ cannot contain two vertices in the same cluster. Hence, $I$ contains one vertex in each cluster of $G$ and does not contain vertices corresponding to complementary literals (i.e. $x_i$ and $\neg x_i$). Thus, it is possible to assign every literal (vertex) in $I$ to be true simultaneously, which constitutes a satisfying assignment for $\phi$.

$\square$

**Hamiltonian Path to Bounded-Degree Spanning Tree**

**Definition 3.** Given an undirected graph $G = (V, E)$, a *Hamiltonian path* is a simple path in $G$ that visits each vertex in $V$ exactly once.

**Problem 3** (Hamiltonian Path)**.**

- **Input:** An undirected graph $G = (V, E)$.
- **Output:** $\begin{cases} 1 & G \text{ has a Hamiltonian path.} \\ 0 & \text{Otherwise.} \end{cases}$

**Definition 4.** Given an undirected graph $G = (V, E)$ and a positive integer $k$, a *degree-$k$ spanning tree* of $G$ is a subgraph $T$ of $G$ such that:

- $T$ is connected;
- $T$ is acyclic;
- $T$ spans all the vertices of $G$ (i.e., includes all vertices in $V$);
- The maximum degree of any vertex in $T$ is at most $k$.

**Problem 4** (Bounded-Degree Spanning Tree)**.**

- **Input:** An undirected graph $G = (V, E)$ and a positive integer $k$.
- **Output:** $\begin{cases} 1 & G \text{ has a degree-}k \text{ spanning tree} \\ 0 & \text{Otherwise} \end{cases}$

**Theorem 3.** *Hamiltonian Path reduces to Bounded-Degree Spanning Tree.*

*Proof.* Assume we have an algorithm $\mathcal{A}$ solving Bounded-Degree Spanning Tree. Then, we can execute the following algorithm to solve Hamiltonian Path:

**Reduction:** Given an instance $G = (V, E)$ of Hamiltonian Path, we construct an instance $(G', k)$ of Bounded-Degree Spanning Tree as follows:

- If $k = 2$, let $G' = G$.
- If $k > 2$:
  - Let $V' = V \cup \{v_1, v_2, \ldots, v_{k-2} \mid v \in V\}$
  - Let $E' = E \cup \{(v, v_i) \mid v \in V, 1 \le i \le k-2\}$

Output the result of $\mathcal{A}$ on $(G', k)$.

**Proof of Correctness:** We claim that $G$ has a Hamiltonian path $\iff$ $G'$ has a degree-$k$ spanning tree. This clearly holds for $k = 2$ as a degree-2 spanning tree is exactly a Hamiltonian path; a tree with maximum degree 2 is a path, and spanning $G$ is equivalent to visiting every vertex. We now show the reduction holds for all $k > 2$:

$\implies$ : Suppose $G$ has a Hamiltonian path $P$. We can construct a degree-$k$ spanning tree $T'$ of $G'$ by taking $P$ and adding all the new edges $(v, v_i)$ for each $v \in V$. This tree spans all vertices of $G'$, is acyclic, and has maximum degree $k$ (2 from the original path plus $k - 2$ new edges).

$\impliedby$ : Conversely, suppose $G'$ has a degree-$k$ spanning tree $T'$. All the new vertices $v_i$ must be leaves in $T'$ as they have degree 1. If we remove these leaves and their incident edges ($k - 2$ per vertex of $G$) from $T'$, we obtain a spanning tree $T$ of $G$ with maximum degree 2, which must be a Hamiltonian path.

$\square$

# E   Examples prompts and responses

This section contains the prompts and responses from the Strawberry model on the *Independent Set to Induced Matching* and *Clique to Balanced Bi-clique* reductions. For the sake of readability, the LaTeXsnippets in the prompts have been made renderable.

**Prompt**

You are a bot designed to write NP-Hardness reductions. You must use the following LaTeX template to write the reduction:

**Theorem 4.** *Problem Independent_Set reduces to Problem Induced_Matching*

*Proof.* Assume we have an algorithm $A$ solving Problem Induced_Matching. Then, we can execute the following algorithm to solve Problem Independent_Set:

**Reduction**  Given inputs $x_1, \cdots$ to Problem Independent_Set, construct an instance of Problem Induced_Matching as follows.

- $y_1 = \cdots$

Output the result of $A$ on $y_1, \cdots$.

**Proof of Correctness**  To establish correctness, it remains to prove that $x_1, \cdots$ contains a foo of size $k \iff y_1, \cdots$ contains a bar of size $k'$.

$\implies$ : Suppose $x_1, \cdots$ contains a foo of size $k$. $\cdots$ Thus, $y_1, \cdots$ contains a bar of size $k'$.

$\impliedby$ : Suppose $y_1, \cdots$ contains a bar of size $k'$. $\cdots$ Thus, $x_1, \cdots$ contains a foo of size $k$. $\cdots$.

$\square$

**Definition 5.** Given an undirected graph $G = (V, E)$, a subset of vertices $S \subseteq V$ is an *independent set* if no two nodes are joined by an edge:

$$\forall u, v \in S : (u, v) \notin E.$$

**Problem 5** (Independent Set)**.**

- **Input:** $(G, k)$ where
    - $G = (V, E)$ is an undirected graph
    - $k$ is a positive integer
- **Output:** $\begin{cases} 1 & G \text{ has an independent set of size } k \\ 0 & \text{otherwise} \end{cases}$

**Definition 6.** Given an undirected graph $G = (V, E)$, a subset of edges $M \subseteq E$ is an *induced matching* if no two edges in $M$ are incident to the same vertex.

**Problem 6** (Induced Matching)**.**

- **Input:** $(G, k)$ where
    - $G = (V, E)$ is an undirected graph
    - $k$ is a positive integer
- **Output:** $\begin{cases} 1 & G \text{ has an induced matching of size } k \\ 0 & \text{otherwise} \end{cases}$

Write a proof that Induced_Matching is NP-hard by reducing from Independent_Set.

**Response**

**Theorem 5.** *Problem Independent Set reduces to Problem Induced Matching*

*Proof.* Assume we have an algorithm $A$ solving Problem Induced Matching. Then, we can execute the following algorithm to solve Problem Independent Set:

**Reduction**  Given inputs $(G, k)$ to Problem Independent Set, construct an instance of Problem Induced Matching as follows.

- For each vertex $v \in V(G)$, create two vertices $u_v$ and $w_v$ in $G'$, and add an edge $(u_v, w_v)$.
- For every edge $(v_i, v_j) \in E(G)$, add edges between all pairs of vertices from $\{u_{v_i}, w_{v_i}\}$ and $\{u_{v_j}, w_{v_j}\}$. Specifically, add edges:
    - $(u_{v_i}, u_{v_j})$
    - $(u_{v_i}, w_{v_j})$
    - $(w_{v_i}, u_{v_j})$
    - $(w_{v_i}, w_{v_j})$

Set $k' = k$.

Output the result of $A$ on $(G', k')$.

**Proof of Correctness** To establish correctness, it remains to prove that $G$ contains an independent set of size $k \iff G'$ contains an induced matching of size $k'$.

$\implies$ : Suppose $G$ contains an independent set $S \subseteq V(G)$ of size $k$. Consider the set of edges $M = \{(u_v, w_v) \mid v \in S\}$ in $G'$.

- Since $S$ is an independent set in $G$, there are no edges between any pair of vertices $v_i, v_j \in S$.
- Therefore, by construction, there are no edges between any pair of edges $(u_{v_i}, w_{v_i})$ and $(u_{v_j}, w_{v_j})$ in $G'$ for $v_i, v_j \in S$.
- Additionally, since each edge in $M$ shares no common vertices with others in $M$, $M$ is a matching in $G'$.
- There are also no edges connecting the endpoints of different edges in $M$ (due to the absence of connections between their vertices), so $M$ is an *induced matching* in $G'$.

Thus, $G'$ contains an induced matching of size $k'$.

$\impliedby$ : Suppose $G'$ contains an induced matching $M$ of size $k' = k$. Let $S = \{v \in V(G) \mid (u_v, w_v) \in M\}$.

- Since $M$ is a matching in $G'$, no two edges in $M$ share a common vertex.
- Since $M$ is an induced matching, there are no edges between the endpoints of different edges in $M$.
- By construction, if there were an edge $(v_i, v_j) \in E(G)$ between any two vertices $v_i, v_j \in S$, then in $G'$, there would be edges between the endpoints of $(u_{v_i}, w_{v_i})$ and $(u_{v_j}, w_{v_j})$, violating the induced matching property.
- Therefore, there are no edges between any pair of vertices in $S$ in $G$.

Thus, $S$ is an independent set of size $k$ in $G$.

Therefore, $G$ contains an independent set of size $k$ if and only if $G'$ contains an induced matching of size $k'$.

$\square$

**Score** This construction does indeed resemble the well-known reduction from Cameron [1989]. The key difference is that Strawberry constructs a clique on the four vertices $\{u_{v_i}, w_{v_i}, u_{v_j}, w_{v_j}\}$ for each edge $(v_i, v_j) \in E$, whereas [Cameron, 1989] constructs a larger gadget, where the four edges between the $u$'s and $w$'s are replaced with disjoint paths of length 8.

Interestingly, a short argument shows that a slight modification of Strawberry's construction does produce a valid reduction. The main point is that $G'$ may contain a matching which contains some "vertical" or "diagonal" edges (i.e., edges not of the form $(u_v, w_v)$). However, it is possible to swap these edges with the desired type of edge, which completes the proof. As a result, this response receives a 1.

**Prompt**

You are a bot designed to write NP-Hardness reductions. You must use the following LaTeX template to write the reduction:

**Theorem 6.** *Problem Clique reduces to Problem Balanced_Biclique*

*Proof.* Assume we have an algorithm $A$ solving Problem Balanced_Biclique. Then, we can execute the following algorithm to solve Problem Clique:

**Reduction** Given inputs $x_1, \cdots$ to Problem Clique, construct an instance of Problem Balanced_Biclique as follows.

- $y_1 = \cdots$

Output the result of $A$ on $y_1, \cdots$.

**Proof of Correctness**  To establish correctness, it remains to prove that $x_1, \cdots$ contains a foo of size $k \iff y_1, \cdots$ contains a bar of size $k'$.

$\implies$ : Suppose $x_1, \cdots$ contains a foo of size $k$. $\cdots$ Thus, $y_1, \cdots$ contains a bar of size $k'$.

$\impliedby$ : Suppose $y_1, \cdots$ contains a bar of size $k'$. $\cdots$ Thus, $x_1, \cdots$ contains a foo of size $k$. $\cdots$.

$\square$

**Definition 7.** Given a undirected graph $G = (V, E)$, a subset of the vertices $S \subseteq V$ is a *clique* if pair of vertices in $S$ is adjacent:

$$\forall u \neq v \in S, (u, v) \in E$$

**Problem 7** (Clique)**.**  The Clique Problem, given a graph $G$ and an integer $k$, is to decide whether or not $G$ contains a clique of size $k$.

- **Input:** $(G, k)$ where $G = (V, E)$ is an undirected graph and $k$ is a positive integer.
- **Output:** $\begin{cases} 1 & G \text{ contains a clique of size at least } k. \\ 0 & \text{Otherwise} \end{cases}$

**Definition 8.** A *balanced bipartite* graph is a bipartite graph in which both sides have the same cardinality; the bipartition $(A, B)$ satisfies $|A| = |B|$.

Given a balanced bipartite graph $G = (V, E)$ with bipartition $(A, B)$, a subset $C \subset V$ is a *balanced biclique* if it is a complete bipartite graph satisfying $|A \cap C| = |B \cap C|$.

**Problem 8** (Balanced Biclique)**.**  The Balanced Biclique Problem, given a balanced bipartite graph $G$ and an integer $k$, is to decide whether or not $G$ contains a balanced biclique of size $k$.

- **Input:** $(G, k)$ where $G = (V, E)$ is a balanced bipartite graph and $k$ is a positive integer.
- **Output:** $\begin{cases} 1 & G \text{ contains a balanced biclique of size at least } k. \\ 0 & \text{Otherwise} \end{cases}$

Write a proof that Balanced_Biclique is NP-hard by reducing from Clique.

**Response**

**Theorem 7.** *Problem* Clique *reduces to Problem* Balanced_Biclique.

*Proof.* Assume we have an algorithm $A$ solving Problem Balanced_Biclique. Then, we can execute the following algorithm to solve Problem Clique:

**Reduction**  Given inputs $(G, k)$ to Problem Clique, construct an instance of Problem Balanced_Biclique as follows.

- Construct a balanced bipartite graph $G' = (V', E')$ with bipartition $(A, B)$ where:
    - $A = V$, the vertex set of $G$.
    - $B = V'$, a copy of the vertex set $V$ (i.e., for each $v \in V$, there is a corresponding $v' \in B$).
    - For each edge $(u, v) \in E$, include edges $(u, v')$ and $(v, u')$ in $E'$, where $u \in A$, $v' \in B$, $v \in A$, and $u' \in B$.
- Set $k' = k$.

Output the result of $A$ on $(G', k')$.

**Proof of Correctness**   To establish correctness, it remains to prove that $G$ contains a clique of size $k \iff G'$ contains a balanced biclique of size $2k$.

$\implies$ : Suppose $G$ contains a clique $C \subseteq V$ of size $k$. Define subsets $A' = C \subseteq A$ and $B' = \{v' \mid v \in C\} \subseteq B$. For every pair $u, v \in C$, since $C$ is a clique in $G$, $(u, v) \in E$. Thus, in $G'$, edges $(u, v')$ and $(v, u')$ are in $E'$. Therefore, every vertex in $A'$ is connected to every vertex in $B'$, forming a balanced biclique of size $2k$ in $G'$.

$\impliedby$ : Suppose $G'$ contains a balanced biclique induced by $A' \subseteq A$ and $B' \subseteq B$, where $|A'| = |B'| = k$, and every vertex in $A'$ is connected to every vertex in $B'$. By the construction of $G'$, an edge $(u, v')$ exists in $E'$ only if $(u, v) \in E$ in $G$. Similarly, since each $v' \in B$ corresponds to a unique $v \in V$. Define $C = A'$. For every pair $u, v \in C$, their corresponding vertices $u' \in B'$, $v' \in B'$ satisfy $(u, v') \in E'$ and $(v, u') \in E'$, implying $(u, v) \in E$ and $(v, u) \in E$ in $G$. Therefore, $C$ is a clique of size $k$ in $G$.

$\square$

**Score**   This response actually contains the well-known "naive" reduction which is somewhat convincing, but completely incorrect. The reason this construction is incorrect is that $G'$ may contain a "misaligned" biclique, where $A'$ and $B'$ do not correspond to the same set of vertices in $G$. For example, take $G = K_{3,3}$ and the construction fails.

## F   List of problems

Table 5: Counts of problem definitions used in reductions

| Problem Name | Source | Dest |
|---|---|---|
| 3 Coloring | 2 | 1 |
| 3D Matching | 6 | 0 |
| 3-Partition | 1 | 1 |
| 3-SAT | 12 | 1 |
| 4 Coloring | 0 | 1 |
| 4D Matching | 0 | 1 |
| 4-Partition | 1 | 1 |
| 4-SAT | 0 | 1 |
| ABCD Partition | 1 | 1 |
| Almost-SAT | 0 | 1 |
| Bin Packing | 0 | 2 |
| Bipartization | 1 | 1 |
| Bounded Degree Spanning Tree | 0 | 1 |
| Clique | 6 | 3 |
| Common Subgraph | 0 | 1 |
| Contagious Set | 0 | 1 |
| Cutting at most K Vertices | 0 | 1 |
| Densest Cut | 0 | 1 |
| Dense Subgraph | 0 | 1 |
| Directed Edge-Disjoint Paths | 0 | 1 |
| Directed Hamiltonian Path | 0 | 1 |
| Dominating Set | 1 | 3 |
| Double-SAT | 0 | 1 |
| Edge Bipartization | 1 | 1 |
| Exact Cover by 3-Sets | 1 | 1 |
| Hamiltonian Cycle | 2 | 0 |
| Hamiltonian Path | 2 | 1 |
| Hitting Set | 0 | 2 |
| Independent Set | 12 | 4 |
| Integer Programming | 0 | 4 |

| | | |
|---|---|---|
| Kernel | 0 | 1 |
| Kite | 0 | 1 |
| Knapsack | 0 | 1 |
| Lecture Planning | 0 | 1 |
| Linear Arrangement | 0 | 1 |
| Longest Path | 0 | 1 |
| Max 2-SAT | 2 | 1 |
| MAX 2-XORSAT | 0 | 1 |
| Max Cover | 0 | 1 |
| Max Cover by Cliques | 0 | 1 |
| Max k-Colorable Subgraph | 0 | 1 |
| Max-SAT | 0 | 1 |
| Min 2-SAT Deletion | 0 | 1 |
| NAE 3SAT | 1 | 1 |
| NAE 4SAT | 1 | 1 |
| Partition | 2 | 1 |
| Path Selection | 0 | 1 |
| Planar 3-Coloring | 0 | 1 |
| SAT | 6 | 0 |
| Set Cover | 5 | 3 |
| Set Packing | 0 | 2 |
| Sparse Subgraph | 0 | 1 |
| Steiner Tree | 0 | 1 |
| Strongly Independent Set | 0 | 2 |
| Subgraph Isomorphism | 1 | 1 |
| Subset Sum | 2 | 2 |
| Suspicious Coalition | 0 | 1 |
| Traveling Salesman | 1 | 1 |
| Triangle Cover | 0 | 2 |
| Undirected Feedback Set | 1 | 2 |
| Unit Intersection | 0 | 1 |
| Unweighted Bisection Width | 0 | 1 |
| Unweighted Max Bisection | 2 | 1 |
| Unweighted Max Cut | 4 | 2 |
| Vertex Cover | 11 | 3 |
| Vertex Disjoint Paths | 0 | 1 |
| Weighted Bisection Width | 0 | 2 |
| Weighted Max Bisection | 1 | 1 |
| Weighted Max Cut | 1 | 0 |
| Zero One Equations | 0 | 1 |
| Zero Weight Cycle | 0 | 1 |

# G   List of reductions

The dataset contains reductions over a wide range of difficulties, from easy generalizations (e.g., SAT to Max-SAT) to complex constructions (e.g., 3-SAT to 3-Coloring). The length of a reduction is a reasonable indicator of its difficulty, so we include the lengths of each reduction (in characters) in the following table. The complete distribution of lengths is visualized in Figure 2.

Table 6: List of reductions between problems

| Source | Destination | Length |
|---|---|---|
| 3-Coloring | 4-Coloring | 1525 |
| 3-Coloring | Planar 3-Coloring | 5789 |
| 3D Matching | 4D Matching | 1701 |
| 3D Matching | ABCD Partition | 3897 |
| 3D Matching | Exact Cover By 3-Sets | 1486 |

| | | |
|---|---|---|
| 3D Matching | Subset Sum | 2331 |
| 3D Matching | Unit Intersection | 1747 |
| 3D Matching | Zero One Equations | 1650 |
| 3-Partition | Bin Packing | 1629 |
| 3-SAT | 3-Coloring | 3553 |
| 3-SAT | 4-SAT | 2130 |
| 3-SAT | Clique | 2337 |
| 3-SAT | Directed Hamiltonian Path | 3680 |
| 3-SAT | Double SAT | 1518 |
| 3-SAT | Independent Set | 2009 |
| 3-SAT | Integer Programming | 2456 |
| 3-SAT | Kernel | 2434 |
| 3-SAT | Max 2-SAT | 2954 |
| 3-SAT | NAE 4-SAT | 1737 |
| 3-SAT | Vertex Cover | 2213 |
| 3-SAT | Vertex Disjoint Paths | 3309 |
| 4-Partition | 3-Partition | 4707 |
| ABCD Partition | 4-Partition | 1821 |
| Bipartization | Vertex Cover | 2692 |
| Clique | Bipartization | 2098 |
| Clique | Cutting At Most K Vertices | 3008 |
| Clique | Dense Subgraph | 1387 |
| Clique | Independent Set | 1505 |
| Clique | KITE | 1443 |
| Clique | Subgraph Isomorphism | 1032 |
| Dominating Set | Set Cover | 1252 |
| Edge Bipartization | Max 2-XORSAT | 2191 |
| Exact Cover By 3-Sets | Steiner Tree | 2066 |
| Hamiltonian Cycle | Hamiltonian Path | 1799 |
| Hamiltonian Cycle | Traveling Salesman | 1425 |
| Hamiltonian Path | Bounded Degree Spanning Tree | 1845 |
| Hamiltonian Path | Longest Path | 967 |
| Independent Set | Clique | 1505 |
| Independent Set | Dominating Set | 3316 |
| Independent Set | Hitting Set | 1463 |
| Independent Set | Integer Programming | 1946 |
| Independent Set | Path Selection | 1821 |
| Independent Set | Set Cover | 1701 |
| Independent Set | Set Packing | 1509 |
| Independent Set | Sparse Subgraph | 1267 |
| Independent Set | Strongly Independent Set | 1944 |
| Independent Set | Triangle Cover | 2631 |
| Independent Set | Undirected Feedback Set | 2398 |
| Independent Set | Vertex Cover | 1308 |
| Max 2-SAT | Min 2-SAT Deletion | 967 |
| Max 2-SAT | Unweighted Max Cut | 4609 |
| NAE 3-SAT | Unweighted Max Cut | 3031 |
| NAE 4-SAT | NAE 3-SAT | 2175 |
| Partition | Bin Packing | 1729 |
| Partition | Knapsack | 1875 |
| SAT | 3-SAT | 2401 |
| SAT | Almost SAT | 1354 |
| SAT | Directed Edge Disjoint Paths | 3925 |
| SAT | Independent Set | 2184 |
| SAT | Max SAT | 1447 |
| SAT | Subset Sum | 3724 |
| Set Cover | Dominating Set | 2115 |
| Set Cover | Integer Programming | 2034 |
| Set Cover | Max Cover | 939 |

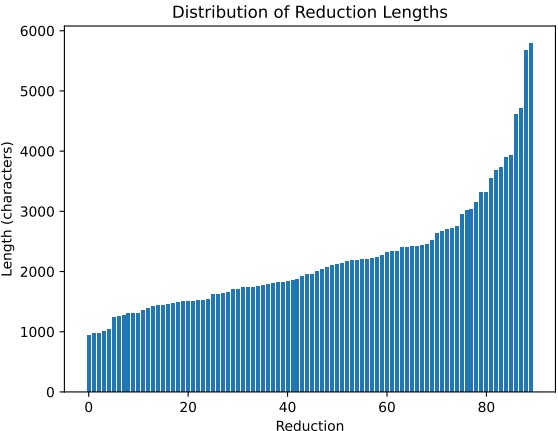

Figure 2: The distribution of lengths (i.e., number of characters) of reductions in the dataset. Most reductions have lengths between 1000 and 3000 characters. The minimum is 939, the maximum is 5789, and the mean is 2180.

| | | |
|---|---|---|
| Set Cover | Max Cover By Cliques | 2755 |
| Set Cover | Max K Colorable Subgraph | 2713 |
| Subgraph Isomorphism | Common Subgraph | 1004 |
| Subset Sum | Partition | 1618 |
| Subset Sum | Zero Weight Cycle | 2312 |
| Traveling Salesman | Integer Programming | 2672 |
| Undirected Feedback Set | Contagious Set | 1790 |
| Unweighted Max Bisection | Unweighted Bisection Width | 1832 |
| Unweighted Max Bisection | Weighted Bisection Width | 2232 |
| Unweighted Max Cut | Densest Cut | 2414 |
| Unweighted Max Cut | Edge Bipartization | 1231 |
| Unweighted Max Cut | Linear Arrangement | 5681 |
| Unweighted Max Cut | Unweighted Max Bisection | 1769 |
| Vertex Cover | Clique | 1431 |
| Vertex Cover | Dominating Set | 3154 |
| Vertex Cover | Hitting Set | 1307 |
| Vertex Cover | Independent Set | 1306 |
| Vertex Cover | Lecture Planning | 1918 |
| Vertex Cover | Set Cover | 1536 |
| Vertex Cover | Set Packing | 1617 |
| Vertex Cover | Strongly Independent Set | 2262 |
| Vertex Cover | Suspicious Coalition | 2194 |
| Vertex Cover | Triangle Cover | 2418 |
| Vertex Cover | Undirected Feedback Set | 2199 |
| Weighted Max Bisection | Weighted Bisection Width | 2509 |
| Weighted Max Cut | Weighted Max Bisection | 1735 |

