# OpenReview forum: "The Karp Dataset"
_NeurIPS.cc/2024/Workshop/MATH-AI — MATH-AI 24_

### Official Review · Reviewer_77Ed · 2024-10-07
**The paper presents a novel Karp dataset for NP-completeness reductions, but its experimental results require more clarity and thorough evaluation.**

**Rating:** 8
**Confidence:** 4

**Review:**

The paper presents the Karp dataset, an important contribution for assessing LLMs on NP-completeness reductions, addressing a critical gap in benchmarking reasoning tasks. The dataset spans simple to complex reductions, enabling robust evaluations. The comparison of state-of-the-art models and the impact of fine-tuning on reasoning performance is a valuable addition. However, the paper could improve by providing a clearer discussion on evaluation metrics and a more detailed error analysis.

---

### Official Review · Reviewer_H6kz · 2024-10-07
**New dataset of detailed proofs of NP-completeness reductions in natural language, with limitations for LLM evaluation**

**Rating:** 6
**Confidence:** 4

**Review:**

Summary:
The paper introduces a new dataset composed of detailed proofs of NP-completeness reductions in natural language.

Strengths:
- This dataset is innovative and appears to be high-quality.

Opportunities for improvements:
- I'm skeptical about the dataset’s utility as a benchmark. As the authors mentioned, testing models on this dataset requires human evaluation, which restricted their tests to only 8 problems. Would it be possible to augment the dataset with formalizations using the Karp programming language, so that models can be tested with an automated verification tool?
- The rating system (0, 1, 2) is somewhat unclear and it might be more intuitive to use a percentage-based system. The paper would also benefit from a clearer description of the dataset, for example by indicating the number of total samples in the dataset, the average length of the proofs, the distribution of the lengths, etc.

Final remark:
The paper introduces a high-quality dataset with potential applications in training/finetuning LLMs. However, due to concerns about its current use for LLM evaluation (reliance on human evaluation), I opt for a rating of 6.

---

### Official Review · Reviewer_CiFv · 2024-10-08
**Significant Contribution to the Field**

**Rating:** 7
**Confidence:** 3

**Review:**

This study presents the creative dataset, which contains detailed proofs of NP-completeness reductions. It serves as a valuable resource for the study and analysis of these complex problems, representing a significant contribution to the field of computational complexity.

The paper's strengths lie in its innovative approach and rigorous experimental design, demonstrating a high level of precision in validating the dataset's effectiveness. However, the study has a few limitations: the range of experiments is somewhat limited, and the validation dataset is relatively small, which could impact the robustness of the results.

Overall, despite these limitations, the paper makes a novel and meaningful contribution to the field, paving the way for further research on NP-completeness and related topics.

---

### Official Review · Reviewer_yfT5 · 2024-10-08
**Review for the Karp Dataset**

**Rating:** 6
**Confidence:** 3

**Review:**

Summary:
The paper introduces the Karp dataset, a structured collection of NP-completeness reductions with varying difficulty designed to test the reasoning ability of language models.

Pros:
- The dataset fills a gap by focusing on reductions, which are not covered by existing math datasets.
- It provides a benchmark for evaluating language models on complex tasks where they currently struggle. Evaluation results show that the dataset can differentiate between models (it would be nice to add more models, e.g., Gemini/Claude).
- Fine-tuning experiments with the dataset show improvements, suggesting the training split could enhance reasoning capabilities of language models.

Cons/Comments:
- On evaluation, manual verification of results can not scale well for larger experiments, which limits the practicality of the evaluation (since grad students in theoretical computer science are becoming scarce these days). This scalability is likely the biggest concern. It would be helpful for the authors to add more discussions on potential strategies to automate this verification process (beyond a formal system, can llm-as-a-judge approach be compared here?) — it should help future researchers to use the dataset more efficiently.
- On fine-tuning LLMs with Karp, it would strengthen the paper if the authors included evaluations on more math or downstream tasks, or providing recipes on mixed dataset training (in Table 4, it looks like that finetuning with Karp can hurt the performance of multilingual GSM8K?). This would give a broader view of how the knowledge contained in dataset improves overall reasoning ability.

Overall, I believe the contribution of this work on the construction of the new dataset is sufficient as an workshop paper.

---

### Official Review · Reviewer_aNtf · 2024-10-09
**review for "The Karp Dataset"**

**Rating:** 6
**Confidence:** 4

**Review:**

Pros:
1. The Karp Dataset is specifically designed to test LLMs on NP-completeness reductions, a challenging task that requires deep mathematical reasoning rather than simple arithmetic or symbolic manipulation. This focus enables a better assessment of an LLM’s ability to handle complex problem-solving tasks, which is a step forward in benchmarking advanced mathematical capabilities.
2. Unlike other math-focused datasets that concentrate on numerical solutions, the Karp Dataset provides a series of reductions involving NP-hard problems. This allows for evaluating and training LLMs in a domain not widely explored by other datasets, helping to fill a gap in the types of problems used for AI evaluation. Such reasoning processes are more advanced and valuable to test.

Cons:
1. While the dataset provides a unique challenge, its focus on NP-completeness may limit its applicability for general mathematical reasoning evaluation. The problems proposed in this paper are very interesting yet still far away from the major scope of math. Yet, this is still a very meaning attempt.
2. The main concern is that this method is hard to extend; it seems that it is not easy to extend such dataset and a small amount of data may results in bias or false statistical significance.

---

### Decision · Program_Chairs · 2024-10-09

Accept